# Machine Learning-Based Radiomics Signatures for EGFR and KRAS Mutations Prediction in Non-Small-Cell Lung Cancer

**DOI:** 10.3390/ijms22179254

**Published:** 2021-08-26

**Authors:** Nguyen Quoc Khanh Le, Quang Hien Kha, Van Hiep Nguyen, Yung-Chieh Chen, Sho-Jen Cheng, Cheng-Yu Chen

**Affiliations:** 1Professional Master Program in Artificial Intelligence in Medicine, College of Medicine, Taipei Medical University, Taipei 106, Taiwan; sandychen@tmu.edu.tw; 2Research Center for Artificial Intelligence in Medicine, Taipei Medical University, Taipei 106, Taiwan; 3Translational Imaging Research Center, Taipei Medical University Hospital, Taipei 110, Taiwan; 4International Master/Ph.D. Program in Medicine, College of Medicine, Taipei Medical University, Taipei 110, Taiwan; m142109004@tmu.edu.tw (Q.H.K.); nguyenhiep0320@gmail.com (V.H.N.); 5Oncology Center, Bai Chay Hospital, Quang Ninh 20000, Vietnam; 6Department of Medical Imaging, Taipei Medical University Hospital, Taipei 11031, Taiwan; rendell2300192@gmail.com; 7Department of Radiology, School of Medicine, College of Medicine, Taipei Medical University, Taipei 11031, Taiwan

**Keywords:** non-small-cell lung carcinoma, EGFR mutation, KRAS mutation, genetic algorithm, eXtreme Gradient Boosting, feature selection, radiogenomics, machine learning, low-dose computed tomography

## Abstract

Early identification of epidermal growth factor receptor (EGFR) and Kirsten rat sarcoma viral oncogene homolog (KRAS) mutations is crucial for selecting a therapeutic strategy for patients with non-small-cell lung cancer (NSCLC). We proposed a machine learning-based model for feature selection and prediction of EGFR and KRAS mutations in patients with NSCLC by including the least number of the most semantic radiomics features. We included a cohort of 161 patients from 211 patients with NSCLC from The Cancer Imaging Archive (TCIA) and analyzed 161 low-dose computed tomography (LDCT) images for detecting EGFR and KRAS mutations. A total of 851 radiomics features, which were classified into 9 categories, were obtained through manual segmentation and radiomics feature extraction from LDCT. We evaluated our models using a validation set consisting of 18 patients derived from the same TCIA dataset. The results showed that the genetic algorithm plus XGBoost classifier exhibited the most favorable performance, with an accuracy of 0.836 and 0.86 for detecting EGFR and KRAS mutations, respectively. We demonstrated that a noninvasive machine learning-based model including the least number of the most semantic radiomics signatures could robustly predict EGFR and KRAS mutations in patients with NSCLC.

## 1. Introduction

Lung cancer is one of the most fatal cancers; it has the second highest prevalence globally, resulting in 18% of cancer cases and 11.4% of deaths globally [1]. Non-small-cell lung cancer (NSCLC) accounts for >85% of reported lung cancers, with adenocarcinoma being the most common pathological type of NSCLC [2]. In 2020, the 5-year overall survival rate of patients with lung cancer was <15%, and 228,820 new cases of lung cancer and 135,720 deaths from lung cancer were estimated to occur [3].

Epidermal growth factor receptor (EGFR) and Kirsten rat sarcoma viral oncogene homolog (KRAS) are the two most frequently observed mutations in patients with NSCLC [4]. Patients with EGFR mutations exhibit higher sensitivity to gefitinib and erlotinib, whereas those with KRAS mutations are prone to drug resistance [4,5,6,7,8,9]. Thus, the early detection of EGFR and KRAS mutations can facilitate the selection of treatment modalities for specific mutation types and enable efficient short- and long-term management, thus improving the prognosis and prolonging the survival duration of patients with NSCLC [4]. Because the number of patients harboring both mutations is considerably limited, the confirmation of one mutation can lead to the exclusion of the other mutation [5]. Although traditional diagnostic methods used to identify mutated sequences in tumor cells have contributed to the advancement of targeted therapies, they are highly invasive and time intensive [9]. Therefore, new noninvasive screening methods should be urgently developed to overcome the drawbacks of invasive traditional approaches [10,11,12,13,14].

With the development of state-of-the-art artificial intelligence techniques, a detailed radiographic quantitative analysis of tumor characteristics can be performed using ubiquitous medical images instead of a qualitative assessment [15]. The aforementioned process is referred to as “radiomics”, which involves the extraction of tumoral features and the evaluation of the association between noninvasive multimodality imaging and pathophysiology [15,16]. Noninvasive models demonstrated a higher predictive power than did routine clinical diagnostic methods [15,17,18,19,20,21,22] and could predict EGFR and KRAS mutations before treatment. In 2017, Gevaert et al. [20] used the decision tree algorithm to categorize the mutations of interest but failed to recognize the KRAS mutation using the algorithm. Furthermore, Shiri et al. [22] established a stochastic gradient descent model with semantic radiographic features and reported that the model exhibited satisfactory performance in identifying patients with NSCLC harboring the EGFR or KRAS mutation. However, the authors’ framework was complex because the model required various radiomics features to achieve high accuracy.

In the present study, we hypothesized that a machine learning-based feature selection and prediction model using the least semantic radiomics features from low-dose computed tomography (CT) findings can predict EGFR and KRAS mutations in patients with NSCLC. We used the public dataset from The Cancer Imaging Archive (TCIA) [23] published by Bakr et al. [24]. The proposed machine learning model is potentially useful for the early prediction of driver gene mutations in patients with NSCLC and thus facilitates early therapeutic intervention and improves overall outcomes.

## 2. Results

### 2.1. Patients’ Characteristics

We retrospectively identified a cohort of 211 patients with NSCLC; of these, 161 with confirmed EGFR and KRAS mutations who met the inclusion criteria [24] were included. Of the 161 patients, 143 (average age: 69.2 ± 8.84 years; 107 men and 36 women) and 18 (average age: 66.9 ± 13.85 years, 4 men and 14 women) were included in the training and validation sets, respectively. According to the exclusion criteria, we excluded 50 patients from the original dataset [24]. Compared with the training set, the validation set consisted of considerably younger patients and a lower proportion of men. No difference in smoking or recurrence status was noted between the training and validation sets. Most of the patients included in the training set had an adenocarcinoma-type tumor (*n* = 111), and only 3 patients had an unspecified histological type. All patients in the validation cohort had adenocarcinoma (*n* = 18). The number of patients with EGFR and KRAS mutations significantly differed between the two cohorts. However, the percentage of patients with cancer recurrence or progression did not significantly differ between the cohorts. No information regarding surgical or posttreatment outcomes was recorded. Table 1 lists the characteristics of patients included in our study.

### 2.2. Radiomics Signature Building

We performed feature selection analysis among different feature categories to investigate and identify crucial feature signatures for our models. The genetic algorithm (GA) [25] exhibited the highest accuracy in predicting both EGFR and KRAS mutations (Figure 1). The number of feature signatures was maintained as low as possible while achieving satisfactory performance.

### 2.3. Supervised Learning Classification

After identifying optimal features using the GA, we examined the performance of different classification algorithms, namely, logistic regression (LR), k-nearest neighbors (kNN), random forest (RF), and eXtreme Gradient Boosting (XGBoost). On the basis of their performance, we subsequently determined the most appropriate algorithm for building the final classification model. In addition, we performed grid-search cross-validation to identify the most optimal parameters for all the aforementioned machine learning algorithms. XGBoost was superior to the other conventional machine learning algorithms in detecting both EGFR and KRAS mutations. As shown in Table 2, our XGBoost model exhibited a sensitivity of 43.5% and 55.6%, specificity of 94.6% and 89.3%, and accuracy of 84.5% and 77.2% in detecting EGFR and KRAS mutations, respectively.

Although the results were satisfactory, an imbalance was observed between sensitivity and specificity. Therefore, we applied the synthetic minority oversampling technique [26] to solve this imbalance problem. As shown in Table 2, we achieved a more balanced result with a sensitivity of 65.2% and 55.6% and a specificity of 88.2% and 95.4% for detecting EGFR and KRAS mutations, respectively. The XGBoost classifier demonstrated the most favorable performance, indicating that it was the optimal algorithm for solving this radiomics-related binomial classification problem.

### 2.4. Validation of Models

We combined the XGBoost classifier algorithm with each optimal feature selector. In addition, we applied our optimal models to validation data to evaluate their performance for unseen data. In this step, our model showed a sensitivity of 66.7% and 33.3%, specificity of 83.3% and 93.3%, and accuracy of 77.8% and 83.3% in detecting EGFR and KRAS mutations, respectively. The sensitivity for the KRAS mutation was lower because of limited data availability (we had only three KRAS mutant samples in the validation cohort, and our model could correctly predict one of them). By contrast, our model could correctly predict various KRAS wild-type samples with an accuracy of 93.3%. This result indicated the efficiency of our model when used for examining different datasets.

### 2.5. Explanation of Feature Selection Using SHAP

Shapley additive explanation (SHAP) analysis [27] was performed to explain the output of our machine learning model and radiomics feature set. As shown in Figure 2, all crucial features were obtained from the wavelet transform. The wavelet-based transform could more effectively improve the radiomics-based prediction model compared with the original radiomics feature. “Energy” was determined to be the most crucial feature; this finding is in accordance with those of previous studies.

Among the seven most favorable features identified in the first model, the wavelet LLL–first-order–energy feature contributed the most to the detection of EGFR status. The first-order statistics refer to the distribution of voxel intensities within the image region defined by the mask through commonly used and basic metrics. Energy is a quantity of voxel values within the regions of interest (ROIs), and a higher energy denotes a larger sum of squares of these values [28].

In terms of the KRAS detection task, the wavelet LLH–gray-level size zone matrix (GLSZM)–large area emphasis was the most appropriate feature. For more information, a GLSZM enumerates gray-level areas (i.e., the number of connected voxels sharing the similar intensity of the gray level in an image), whereas large area emphasis (LAE) measures the distribution of regions with a large area; a higher LAE value indicates the presence of greater size zones and more coarse textures [28].

### 2.6. Comparison with Previous Radiomics-Based EGFR and KRAS Prediction Models

We compared the predictive performance of our model with that of radiomics-based models used in previous studies for predicting EGFR and KRAS mutations. Pinheiro et al. [29], Shiri et al. [22], and Zhang et al. [30] used the same dataset as we did; thus, we compared their results with our findings. A detailed comparison is presented in Table 3 and Section 3.

## 3. Discussion

In our study, we proposed a comprehensive machine learning approach and feature selection assessment to efficiently detect EGFR and KRAS mutations in patients with NSCLC; early detection of these mutations can help in the administration of suitable individualized and targeted treatment options. The public dataset consisting of 211 patients with NSCLC was derived from a previous study on the TCIA [23,24]. Of these, 143 patients (116 EGFR and 114 KRAS mutation carriers) and 18 patients (6 EGFR and 3 KRAS mutation carriers) were included in the training and validation cohorts, respectively.

Our proposed model correctly predicted EGFR and KRAS mutations with an area under the curve (AUC) of 0.89 and 0.812, respectively. According to the results, it is observed that there were about 10% of examined patients that had false negatives or positives. To explain this, our strategy failed in those cases because of the limit of training data. The model might not have enough data to train sufficiently, and it has failed in some cases that they did not learn well. In the future, more comprehensive data should be retrieved to reach a better performance with less false positives and negatives. The AUC of the KRAS model in the validation set was lower than that in the other model due to imbalanced data; only 3 of 18 patients had the KRAS mutation. However, the specificity of this model was remarkably high (95.4%), indicating that our second model could still perform efficiently on different datasets regardless of their distribution. Moreover, we achieved these results by including only 7 and 11 features in the EGFR and KRAS model, respectively.

Pinheiro et al. [29] and Shiri et al. [22] have analyzed and interpreted radiomics features extracted from the same public dataset included in the current study [23,24]. Pinheiro et al. introduced a model built on the XGBoost algorithm and its feature importance ranking function. Utilizing the similar classifier to predict EGFR mutations, the authors selected a subset of the 37 most semantic radiomics features, yielding an AUC of 0.7458 (vs. 0.89 in our model); however, the model failed to detect the KRAS mutation with an AUC of 0.5035 (vs. 0.812 in our model). The decreased efficiency of the model in classifying EGFR and KRAS mutations [29] might be due to differences in feature selection methods. Despite the use of the same classifier algorithm, the authors employed Pearson’s correlation coefficient method to exclude the least significant features. The radiomics features are continuous numerical variables that may contain outliers, resulting in a low or even insignificant correlation coefficient between features [31].

The model developed by Shiri et al. [22] demonstrated more favorable performance in detecting the KRAS mutation. Their reported AUC of 0.83 was slightly higher than our AUC of 0.812. However, the detection of the EGFR mutation did not yield a similar outcome; our EGFR model could generate a superior result to the “stochastic gradient boosting–select from model” ensemble, with AUCs of 0.89 and 0.78 for our study and previous study, respectively. Their results were not as distinct as those of our model. Therefore, our model exhibited relatively stable performance in the detection of both mutation types.

Zhang et al. [30] conducted a retrospective single-center study of 248 patients with lung adenocarcinoma (135 men and 113 women, mean age of 62.43 ± 9.19 years) and investigated the prediction capacity of pretherapy 18F-FDG PET/CT-based radiomics features for EGFR mutation status in patients with NSCLC. All patients were histologically confirmed to have the EGFR mutation (i.e., mutant or wild-type) and underwent 18F-FDG PET/CT 1 month prior to treatment. The original dataset was then randomly split into training and validation sets containing 175 and 73 patients, respectively. The LIFEx package [32] was used to extract 47 PET radiomics features and 45 CT radiomics features in combination with the segmentation of ROIs by two radiologists. The 10 most semantic predictive features were retained using the Mann–Whitney U test and the “least absolute shrinkage and selection operator” regression analysis (10-fold cross-validation), and the establishment of the model was based on the LR algorithm to measure the “rad-score” for each patient. In terms of the predictive efficacy of the EGFR mutation, the model containing only signature radiomics features or clinical features demonstrated a higher AUC than did our proposed model (i.e., 0.85 vs. 0.845), whereas the AUC generated from the clinical model was significantly lower than that of our model (i.e., 0.78 vs. 0.845). Considering the higher sensitivity (91.7 vs. 69.6) and lower specificity (70.3 vs. 83.9), the rad-score and clinical complex model demonstrated, to some extent, a lower accuracy (80.8 vs. 81.0) but a significantly higher AUC (0.87 vs. 0.845). The identification of EGFR mutants is relevant to clinical practice because it can help determine the therapeutic strategy for specific individuals carrying mutations associated with improved sensitivity to gefitinib and erlotinib [4,5,7]. In this study, with the inclusion of only 6 EGFR mutation carriers in the validation set compared with 12 EGFR wild-type carriers, the sensitivity score was acceptable. For the subsequent steps, we boosted our model to increase sensitivity and detect the mutants more efficiently; however, with only seven semantic radiomics features, the EGFR model could predict imbalanced data, thus demonstrating itself as a valuable predictive model for further studies.

Although the performance of our models was considerably promising, our study has various limitations that should be addressed. First, the original dataset [23,24] consisted of 211 patients; however, only 161 patients were included, and no external validation set was considered. Although the results were favorable, we must still evaluate the predictive performance of our two models by including a broader cohort of patients with NSCLC with known EGFR and KRAS mutation status. Second, we must advance the model’s capacity for handling imbalanced data, thereby generalizing the prediction outcome to more datasets. Third, our study could apply the state-of-the-art deep learning algorithm to implement the classification task. Many studies have successfully developed models to solve this problem [30,33,34,35,36] with satisfactory results. These findings encourage us to conduct future studies using neural networks to build the baseline model. Finally, the radiogenomics model can be considered by applying it to more comprehensive genotypes such as EGFR-TKI sensitivity [37] or exon levels [38]. Thereafter, it can be applied for clinical settings in the future.

## 4. Materials and Methods

Figure 3 presents our proposed radiogenomics framework for predicting EGFR and KRAS mutations in patients with NSCLC.

### 4.1. NSCLC Patient Cohort

Datasets used in this study were retrieved from TCIA [23], which is a comprehensive resource for cancer imaging data. Because we aimed to improve the prediction of EGFR and KRAS mutations, we used a benchmark dataset that was introduced by [24]. This dataset comprises the CT images, positron emission tomography (PET)/CT images, and semantic annotations of tumors for a total of 211 patients. This dataset included two cohorts: the R01 cohort consisted of 162 patients with NSCLC, and the AMC cohort consisted of 49 additional patients. The first cohort was retrieved from Stanford University School of Medicine and Palo Alto Veterans Affairs Healthcare System, whereas the second cohort was retrieved only from Stanford University School of Medicine at different timelines. For both cohorts, the following clinical data were obtained where available: smoking history (211), survival (211), recurrence status (210), histology (211), histopathological grading (162), and pathological TNM staging (161). We used R01 as the training cohort and AMC as the validation cohort to examine the performance of the model.

Only patients who met the inclusion criteria were enrolled in the final dataset. The inclusion criteria for patients were as follows: (i) availability of images with less or no noise phenomenon, (ii) absence of artificial images, and (iii) availability of accurately segmented and full-sequence images. Finally, we included 161 patients in our dataset, of whom 143 and 18 were included in the training and validation sets, respectively. Since two datasets came from two different cohorts, the validation data can be treated as external data to evaluate the performance of the model.

### 4.2. Segmentation of Lung Tumors

We analyzed the CT findings of 161 patients from the dataset [24] who met the inclusion criteria. The ROIs were manually segmented by a chest radiologist with 8 years of experience and edited, where necessary, using ePAD [39]. Subsequently, all ROIs manually segmented by the first radiologist were evaluated by the second radiologist with 10 years of experience. Some unreasonable segmentations for the boundaries of ROIs were modified, where appropriate, to ensure segmentation precision.

### 4.3. Radiomics Feature Extraction

In this study, lung tumors were manually segmented by two doctors with 8–10 years of work experience. Subsequently, we used PyRadiomics library [28] to extract the different characteristics of radiomics features, namely, (i) intensity, (ii) image derivatives, (iii) geodesic information, (iv) texture features, and (v) posterior probability maps. In total, 851 features extracted from the library were categorized into 9 classes, namely, original, wavelet HHH, wavelet HHL, wavelet HLH, wavelet HLL, wavelet LHH, wavelet LHL, wavelet LLH, and wavelet LLL. Each category consisted of six subcategories, namely, first-order, gray-level co-occurrence matrix, GLSZM, gray-level run length matrix, neighboring gray tone difference matrix, and gray-level dependence matrix. The original radiomics category consisted of one additional subcategory, namely, shape. The radiomics classes were described by van Griethuysen et al. [28]. Because we would like to assess the performance results using radiomics, we did not include any pathological features in the models.

### 4.4. Feature Selection

We performed different feature selection analyses to identify features that might be crucial for our model. We used the following feature selection techniques: univariate selection, recursive feature elimination (RFE), feature importance, filter methods, F-score, GA, minimum redundancy feature selection, and the KBest algorithm.

The advantages and disadvantages of each technique are described as follows.

#### 4.4.1. Univariate Selection

Univariate feature selection [40] is performed to investigate each separated feature to determine the magnitude of the association between features and the target variable. With various options available, this method can efficiently manipulate data.

#### 4.4.2. RFE

RFE [41] involves the removal of the most insignificant features until the model has the optimal number of features. Moreover, RFE aims to exclude features, thus resulting in collinearity.

This technique optimizes the number of features remaining in the final model using cross-validation scores to distinguish data subsets. In addition, RFE can be used to visualize chosen features through graphs accompanied by their respective scores and to examine the contribution of the features to the predictive outcome.

#### 4.4.3. Feature Importance

Feature importance [42] assesses the contribution of each feature included in the model on the basis of their assigned scores obtained from regression analysis or classification. In this study, feature importance was used to identify features that were the most efficient in classifying EGFR and KRAS mutations in patients with NSCLC. Thus, feature importance can provide better insights into the model and more semantic interpretations.

#### 4.4.4. Filter Methods

Filter methods [43] refer to a group of various statistical methods primarily used for data preprocessing. Four main statistical tests are used depending on the types of input features and the output variable, namely, Pearson’s correlation coefficient, linear discriminant analysis, analysis of variance, and a chi-squared test. Similar to other feature selection methods, filter methods select features based on scores that manifest the strength of the correlation or association of inspected input features with the predictive output. However, filter methods cannot remove or reduce the multicollinearity phenomenon, which can deteriorate the comprehensive performance of the model.

#### 4.4.5. F-Score

F-score is a feature selection method for examining the difference between two datasets containing continuous numerical values [44]. In the binary classification task, the F-scores of related features are calculated based on Equation (1). In this study, we calculated the F-score to obtain optimal features.
(1)$$F\left(i\right)\equiv \frac{{\left({\bar{𝕩}}_{i}{}^{\left(+\right)}-{\bar{𝕩}}_{i}\right)}^{2}+{\left({\bar{𝕩}}_{i}{}^{\left(-\right)}-{\bar{𝕩}}_{i}\right)}^{2}}{\frac{1}{n_{+}-1}\sum_{k=1}^{n_{+}}{\left(x_{k,i}^{\left(+\right)}-{\bar{𝕩}}_{i}{}^{\left(+\right)}\right)}^{2}+\frac{1}{n_{-}-1}\sum_{k=1}^{n_{-}}{\left(x_{k,i}^{\left(-\right)}-{\bar{𝕩}}_{i}{}^{\left(-\right)}\right)}^{2}}$$
where ${\bar{𝕩}}_{i}{}^{\left(+\right)},{\bar{𝕩}}_{i}{}^{\left(-\right)},\;{\bar{𝕩}}_{i},\;i$ are the mean ith feature of the whole, positive, and negative datasets, respectively, $x_{k,i}^{\left(+\right)}$ is the ith feature of the kth positive instance, and $x_{k,i}^{\left(-\right)}$ is the ith feature of the *k*th negative instance. The numerator distinguishes between positive and negative sets, and the denominator interprets the one within each of the two sets. The higher the F-score is, the more likely the feature is to be more distinctive, thus making it more prominent compared with other features. Although the F-score cannot provide common information among discriminative features, it is a simple yet effective feature selection method for dichotomous problems.

#### 4.4.6. GA

The GA, which is a stochastic feature selection algorithm based on the principle of natural selection, optimizes model performance by identifying the most favorable features [25]. The method produces numerous virtual generations, and each generation contains many individual features that are best suited to the training model and can yield the highest accuracy.

#### 4.4.7. Minimum Redundancy Feature Selection

This method investigates the correlation between features and the predictive outcome and between the features themselves [45]. In the F-statistic test, the level of the correlation between the output and features must be high (relevance), whereas the correlation between features must be low (redundancy). Thereafter, the algorithm filters out a subset of features satisfying both the aforementioned criteria. Two objective functions, namely, the mutual information difference and mutual information quotient, are used to measure the discrimination or the level of relevance and redundancy, respectively [45].

#### 4.4.8. KBest Algorithm

The features are selected based on their importance scores for the target variable. Subsequently, k most significant features are selected.

### 4.5. Machine Learning

Machine learning, which is based on artificial intelligence, involves the use of various algorithms that facilitate the extraction of semantic radiomics features from arbitrary medical imaging datasets [46,47]. Recent studies have demonstrated the capability of machine learning techniques to predict tumor classification and survival rates and even perform molecular profiling based on imaging biomarkers [46,48,49]. In the current study, we evaluated the predictive potential of four machine learning algorithms for the classification task. Among four different classifiers, the kNN and LR algorithms follow the principle of distance function learning and the logistic theorem, respectively. Although both RF and XGBoost are ensemble learning methods based on the votes of individual decision trees, XGBoost can handle imbalanced dichotomous data and visualize the ranking of feature importance for more favorable outcome interpretation [50]. For each of the aforementioned algorithms, we performed cross-validation to identify optimal parameters.

We used machine learning algorithms to classify and robustly predict EGFR and KRAS mutations in patients with NSCLC to improve their overall prognosis and treatment response.

### 4.6. Statistical Analysis

Student’s *t* test and the Mann–Whitney U test were performed to compare continuous and categorical variables, respectively, between the training and validation cohorts. All statistical analyses were conducted using Python. To determine the model that exhibited the most favorable performance in the binomial classification problem, we validated the performance of each model by examining its sensitivity, specificity, and accuracy as follows:(2)$$Sensitivity=\frac{TP}{TP+FN}$$
(3)$$Specificity=\frac{TN}{TN+FP}$$
(4)$$Accuracy=\frac{TP+TN}{TP+FP+TN+FN}$$
where *TP, TN, FP*, and *FN* represent the true positive, true negative, false positive, and false negative, respectively. 

To better visualize and compare the performance of four algorithms and feature selection methods, the receiver operating characteristic curves were plotted using comparable AUC values.

## 5. Conclusions

In this study, we proposed a comprehensive, noninvasive machine learning approach utilizing the classifier algorithms and feature selection methods to robustly predict EGFR and KRAS mutations among patients with NSCLC based on medical radiomics features. We demonstrated that by using a subset of the least number of semantic features, we could reach a promising performance with AUC of 0.89 and 0.812 for EGFR and KRAS mutation status prediction, respectively. The established radiomics signature model helps to accelerate the diagnosis of mutations of interest, thus improving the outcomes of patients with NSCLC in the future. Furthermore, our radiogenomics framework holds potential in solving extension problems in NSCLC including other mutations or exon levels, and then provides more outcomes.

## Figures and Tables

**Figure 1 ijms-22-09254-f001:**
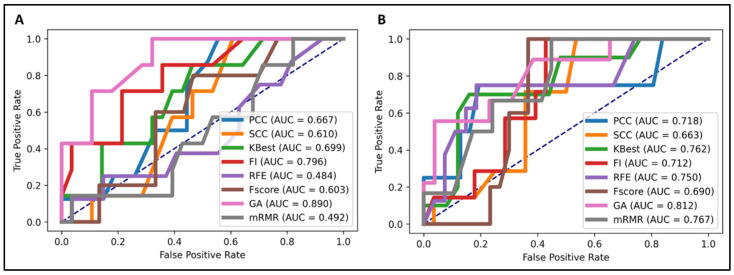
Predictive performance of different feature selection techniques in the NSCLC radiogenomics model. The genetic algorithm (GA) demonstrated the most favorable performance. (**A**) EGFR mutation prediction (area under the curve (AUC) = 0.89), (**B**) KRAS mutation prediction (AUC = 0.812).

**Figure 2 ijms-22-09254-f002:**
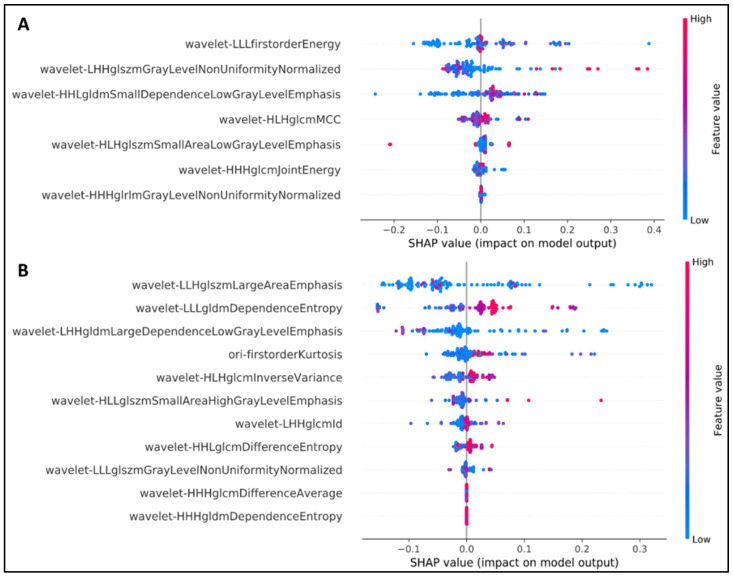
SHAP analysis to explain the crucial features of our NSCLC radiogenomics models. (**A**) EGFR mutation status, (**B**) KRAS mutation status.

**Figure 3 ijms-22-09254-f003:**
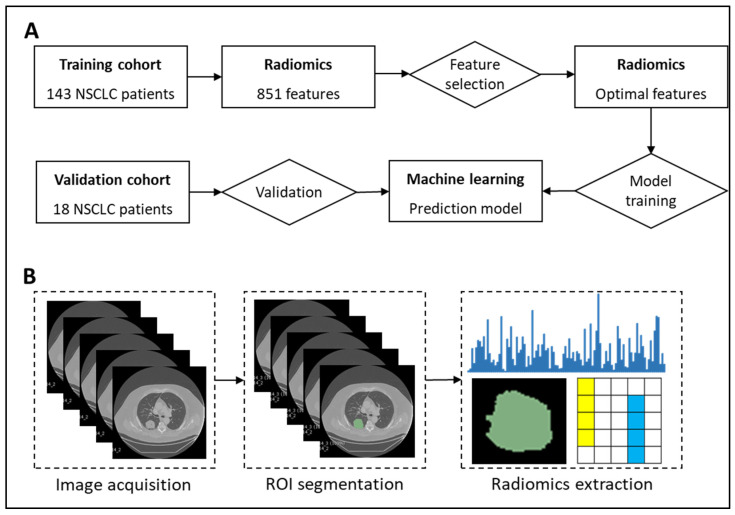
Prediction of EGFR mutation using radiomics, feature selection, and machine learning algorithms. (**A**) Study framework. (**B**) Workflow of radiomics.

**Table 1 ijms-22-09254-t001:** Characteristics of patients in our cohorts.

	Training (*n* = 143)	Validation (*n* = 18)	*p* Value
Age (mean ± SD, years)	69.2 ± 8.84	66.9 ± 13.85	0.334
Sex			0.123
Male	107	4	
Female	36	14	
Smoking status			0.069
Current	30		
Former	91	9	
Nonsmoker	22	9	
Histological type			0.5
Adenocarcinoma	111	18	
NSCLC NOS ^1^	3	0	
Squamous cell carcinoma	29	0	
EGFR mutation			0.074
Mutant	23	6	
Wild-type	93	12	
KRAS mutation			0.074
Mutant	27	3	
Wild-type	87	15	
Recurrence			0.123
Yes	40	3	
No	103	15	

^1^ No other specific type.

**Table 2 ijms-22-09254-t002:** Assessment of different feature selection-based machine learning models for predicting EGFR and KRAS mutations.

		Original			SMOTE		
		Sens	Spec	Acc	Sens	Spec	Acc
EGFR	LR	4.3	100	81	43.5	78.5	71.6
	kNN	34.8	92.5	81	60.9	67.7	66.4
	RF	21.7	97.8	82.8	52.2	84.9	78.4
	XGBoost	43.5	94.6	84.5	65.2	88.2	83.6
KRAS	LR	11.1	98.9	78.1	48.1	73.6	67.5
	kNN	18.5	98.9	79.8	55.6	67.8	64.9
	RF	33.3	96.6	81.6	51.9	75.9	70.2
	XGBoost	55.6	89.3	77.2	55.6	95.4	86

LR: logistic regression, kNN: k-nearest neighbors, RF: random forest, XGBoost: eXtreme Gradient Boosting.

**Table 3 ijms-22-09254-t003:** Comparison between our proposed models and those of previous studies using the same TCIA data.

		Sens	Spec	Acc	AUC
EGFR	Pinheiro et al.	-	-	-	0.7458
	Shiri et al.	-	-	-	0.78
	Zhang et al.	91.7	70.3	80.8	0.87
	Ours	65.2	88.2	83.6	0.89
KRAS	Pinheiro et al.	11.1	98.9	78.1	0.5035
	Shiri et al.	18.5	98.9	79.8	0.83
	Ours	55.6	95.4	86	0.812

- means the metrics were not reported in the corresponding papers.

## Data Availability

Public data can be freely accessed and downloaded at https://wiki.cancerimagingarchive.net/display/Public/NSCLC+Radiogenomics (accessed on 6 June 2020).

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
