# Peer review of "Machine Learning-Based Radiomics Signatures for EGFR and KRAS Mutations Prediction in Non-Small-Cell Lung Cancer"

_ijms, 2021, doi:10.3390/ijms22179254_

Round 1

Reviewer 1 Report

The study is interesting and promising for the NSLC patients The conclusion should be improved to address the main results of the study.

Reviewer 2 Report

Some papers about radiomics were published already. However the scientific knowelege is evolving. Unlike the description in the introduction, it is not difficult to find molecular abnormalities from patients. Because noninvasive technogoly such as plasma tumor DNA could reveal the nature of molecular abnormalities. And recent advance in research shows that even G12C KRAS mutation is also targeted with drug such as sotoracib. 

Author Response

We greatly appreciate the reviewer #2 to carefully review the manuscript and offer positive comments on our work. In the revised manuscript, we have added more discussions and conclusion to provide detailed information for readers. We feel that the revised manuscript is of considerably improved quality and hope that it can meet the increasingly high-quality standard of this journal.

Reviewer 3 Report

The authors have established machine learning-based radiomics signatures for EGFR and KRAS mutations prediction in NSCLC. Their strategy would be very interesting, however over 10% of examined patients had false negative or positive. 

Major revision

  • In false cases, why did their strategy fail?
  • In future, did they apply for clinical setting? Did they predict the clinical outcome including EGFR-TKI sensitivity?
  • How did they select 18 patients as validation study without bias?
  • Do their method correlate pathological feature?
  • Does the ability of this method change in clinical staging?

Author Response

We greatly appreciate the reviewer #3 to carefully review the manuscript and offer valuable suggestions on our work. All of the comments and suggestions were useful for us to improve the quality of this manuscript. We have carefully addressed all fruitful comments in the below section. We feel that the revised manuscript is of considerably improved quality and hope that it can meet the increasingly high-quality standard of this journal.

Comment #1: In false cases, why did their strategy fail?

Answer:

We greatly appreciate the reviewer for fruitful comment. In the revised version, we have discussed some probabilities that might affect the false positive and negative cases as follows: (Section 3, page 6)

“Our proposed model correctly predicted EGFR and KRAS mutations with an area under the curve (AUC) of 0.89 and 0.812, respectively. According to the results, it is observed that there were about 10% of examined patients had false negative or positive. To explain this, our strategy failed in those cases because of the limit of training data. The model might not have enough data to train sufficiently and it has failed in some cases that they did not learn well. In the future, a more comprehensive data should be retrieved to reach a better performance with less false positive and negative.”

Comment #2: In future, did they apply for clinical setting? Did they predict the clinical outcome including EGFR-TKI sensitivity?

Answer:

It should be possible to use our method to predict other clinical outcomes and apply for clinical setting. Thus in the revised version, we have added more discussions to show some potentials in the future to apply the method for other clinical outcomes as follows: (Section 3, page 8)

“Finally, the radiogenomics model can be considered applying to more comprehensive genotypes such as EGFR-TKI sensitivity [37] or exon levels [38]. Thereafter, it can be applied for clinical setting in the future.”

Comment #3: How did they select 18 patients as validation study without bias?

Answer:

We greatly appreciate the reviewer for fruitful comment. The reason why we selected 18 patients in validation study is that it is from another patient cohort (different with training cohort). It is not a random pick and this selection ensures we can apply our training model on different dataset. We have clearly explained this part in the revised manuscript: (Section 4.1, page 9)

“Since two datasets came from two different cohorts, the validation data can be treated as an external data to evaluate the performance of model.”

Comment #4: Do their method correlate pathological feature?

Answer:

Since we would like to evaluate the potential of only radiomics features in predicting EGFR and KRAS mutation statuses, we did not include any pathological feature in our method. We have described this part clearly in the revised manuscript: (Section 4.3, page 9)

“Because we would like to assess the performance results using radiomics, we did not include any pathological features in the models.”

Comment #5: Does the ability of this method change in clinical staging?

Answer:

We greatly appreciate the reviewer for fruitful comment. Because we trained the model using the patient data from different clinical stages of NSCLC, it is designed for prediction of genomics mutation status in all stages. Therefore, the method will not be changed in clinical staging.

Round 2

Reviewer 2 Report

The manuscript was written well scientifically. But as a clinician, I think we do not need to depend on these facts for treatment decision as we can get molecular information more easily from blood or tumor tissue.

Reviewer 3 Report

This manuscript would be acceptable without further revision.